# The Development of a Built-In Shoe Plantar Pressure Measurement System for Children

**DOI:** 10.3390/s22218327

**Published:** 2022-10-30

**Authors:** Sarah De Guzman, Andrew Lowe, Cylie Williams, Anubha Kalra, Gautam Anand

**Affiliations:** 1School of Engineering, Computing and Mathematical Sciences, Auckland University of Technology, Auckland 1010, New Zealand; 2Faculty of Medicine, Nursing and Health Sciences, Monash University, Sydney 2109, Australia

**Keywords:** sensor, foot, child, toddler

## Abstract

There is a rapid increase in plantar pressure from the infant to toddler stage, yet little is known about the reasons for this change. More information about plantar pressure distribution can help clinicians identify early-stage foot-related diseases that may occur during transitions from childhood to adulthood. This information also helps designers create shoes that adapt to different needs. This research describes the development of a low-cost, built-in shoe plantar pressure measurement system that determines foot pressure distribution in toddlers. The study aimed to improve and provide data on pressure distribution during foot growth. This was accomplished by implementing a plantar pressure capacitive measurement system within shoes. The capacitive sensors were laminated using a copper tape sheet on plastic backing with adhesive, elastomer layers, and a combination of conductive and non-conductive fabrics. Constructed sensors were characterized using compression tests with repeated loads. Results demonstrated that the sensors exhibited rate-independent hysteresis in the estimation of pressure. This enabled a calibration model to be developed. The system can mimic more expensive plantar pressure measurement systems at lower fidelity. This emerging technology could be utilized to aid clinicians, researchers, and footwear designers interested in how pressure distribution changes from infants to toddlers.

## 1. Introduction

Sensor technology in the medical and health industry has been increasing in demand with the evolution of mobile and home IoT health devices providing users remote analytics regarding the state of their health. Such examples of this technology are wristband heart rate monitors, glucose monitoring systems for diabetics, as well as sleep inspecting devices.

Children’s foot pressure is highly dynamic due to growth, weight change, and skill demands [1]. At birth, some bones are present but the majority of the foot is cartligenious, transitioning with rapid ossification in the first 12 months to 6 years [2]. A large fat pad is present underneath the foot up to the age of six. It is thought this structure distributes plantar pressures to protect delicate soft cartilage and tissues and help increase the surface contact with the ground during learning complex walking and running skills [3].

One way children’s feet are assessed in clinical environments is through measuring and understanding dynamic plantar pressure patterns [4]. It is not typical that young children to have regular foot assessments [5] unless they have complex health conditions [6,7]. Due to this, there is little normative information about pressure distribution and changes for young children. This lack of knowledge leads to limited information on the variation in foot growth during early childhood, the impact of the different activities on change in pressure distribution, or for other industries such as footwear designers. 

This means there is a need for an accessible database of pressures for all ages. Other known databases aid in understanding the variation during aging, however, these only start from the age of three and do not include plantar pressures [8]. Bosch et al. [9] commenced curating normative values for plantar pressure data for 100 typically developing children over four years, as no information existed to determine the normality of an individual pressure pattern. Their findings provide valuable information about normal growth-related changes in foot shape and foot loading parameters (about plantar pressure). However, their information was over a limited timeframe and with limited appreciation and description for foot posture. Even with this information, there is limited transferable science from clinical research into children’s footwear science to advance understanding of the interaction between the foot and the shoe in young children [10]. There are a small number of existing studies that discuss how shoes affect young children’s gait [11,12,13]. Extremely soft-soled footwear has shown a limited effect on gait in a small number of toddlers [13]. Whereas footwear can increase velocity, step-time, and step length compared to barefoot walking in children [11,12,14]. Only one study has examined the differences in shoe flexibility on plantar pressure loading, founding a very flexible shoe showed the same plantar pressure as a barefoot pressure reading, whereas medium to low flexibility shoes gave lower plantar peak pressures [15].

Finding these differences between different shoe soles is important. However, how data has been collected is limited by age and expensive technology. Capacitive pressure sensors have been reliably used in various technologies such as touch screen devices, and digital audio players, as well as a substitute for mechanical switches. This means, there is potential for a suitable capacitive sensor to be placed within children’s shoes, therefore identifying changes and patterns during foot growth. This simple method of data collection may assist in normative research dataset development or during monitoring of children with conditions known to impact foot health.

This study aimed to develop and validate an in-shoe monitoring tool using a low-cost sensor assembly that mimics pressure sensing insoles currently available on the market. 

## 2. Materials and Methods

The overall system contains the following four components as shown in Figure 1: (a)Mechanical/Physical Sensor assembly—This includes the sensor pads that mainly detect the pressure change.(b)Connections and Routing system—This includes the connection from the sensor pads to the electronic system.(c)Electronics System—This includes the implementation of Arduino 101 and MPR121 which reads the capacitance of the sensor.(d)Data Visualisation and Data Logger—This includes LabVIEW Setup for visualization and logging of data for post-processing.

The proposed built-in-shoe foot pressure measurement system was developed through the following approach: (a)A qualitative analysis of existing manufacturing techniques was conducted on the fabrication of the plantar pressure sensor. A technique that would be inexpensive and still deliver a high-quality sensor was chosen.(b)Testing, Validation, and Calibration—After the sensor was manufactured, it was tested to see whether any connection faults were present. Data was logged and visualized using LabVIEW after the plantar pressure sensor was connected to Arduino and the MPR121 electronics unit. This test was referred to as Validation Testing 1 (VT1). Following calibration curves derived from the plantar pressure sensor system, calibration was performed. This was achieved with the repeated loading of the sensor using controlled instruments. Hounsfield H10KS Tensile Testing machine was configured as a compression testing machine to apply loads to the sensor. This stage was referred to as Validation Testing 2 (VT2).

### 2.1. Sensor Design Layout

This included the mechanical design and development of the sensor assembly comprising of the development of sensor topography/shapes and location, evaluation of materials and manufacturing techniques suitable for the creation of the sensors, and the assembly of the sensors. The plantar pressure sensor was aimed to measure the overall plantar pressure. To do this, the sensor needed to be fixed in a location where it is going to experience minimal movement during the child’s dynamic activity. The plantar pressure sensor was placed in the Strobel area, located between the insole and the midsole of the shoes. Herein, the sensor was fixed (i.e., not loose) preventing it from moving around. Having the sensor fixed in place decreases the amount of environmental noise hence a cleaner signal. This area was also ideal as the sensors can be layered with other protective materials, such as EVA and other lining materials. In this work, the sensor was not tested while it is inside a shoe. Instead, the characteristics of the layers on top of the sensor are observed and characterized. 

Pressure points of interest were informed by the patterns collected by researchers and clinicians using gold standard devices. These devices contain an array of sensors to capture all the pressure points of the plantar area and they have their own software. Masking is one of the functionalities in the accompanying software. Masking considers where plantar areas can be divided up to observe areas of interest. The masks can be setup differently depending on clinician’s and researcher’s interests.

### 2.2. Sensor Manufacturing and Routing

The sensor pads need to be connected to the main electronics unit so that they can be used as pressure sensors. Hence, different manufacturing techniques were explored and qualitatively evaluated. Some techniques are readily available, and these techniques were applied in making the sensors [16,17]. The physical sensors were visually assessed, and their effectiveness was tested by connecting the assembled sensors to MPR121 and testing whether the capacitance would change with a fingertip touch. Vinyl cutting is one of the techniques where there is a computerized blade that can cut the pattern. 

The cut-out of the sensor areas shown in Figure 2 was done using a vinyl cutter, Roland CAMM-1 GS-24.

The sheet-fed on the vinyl cutter was a copper tape sheet with plastic backing. The vinyl cutter is set up to cut with the right amount of force so that it only cuts through the copper tape and not the plastic backing. The unnecessary areas of copper tape are weeded away only keeping the wanted sensor areas. The sensor route layer was also cut using the vinyl cutting machine. These are on a separate sheet for the sensor pads. The sensor route layer and the sensor pads are connected by placing a conductive thread (Figure 3) on the sensor route and this is fed through the through hole (Figure 4) of the sensor pad sheet.

The thread fed through the hole is secured with conductive tape to ensure contact with the sensor pads (Figure 5 and Figure 6).

### 2.3. Sensor Layer Layout

The sensor layer layout consists of the manufactured sensor pads and route layer using the vinyl cutting machine as pictured in Figure 7 and Figure 8.

The functions of these layers were as follows:Cotton Cloth–Sock liner—This served as the protection of the copper taffeta from the environment (e.g., sweaty skin). The copper taffeta tends to tarnish with moisture contact. This also functions as an insulator of the ground plane.Copper Taffeta–Ground Planes—These are present in both the top and bottom layers of the sensor. These reduce the amount of interference from the environment (such Figure 7: Side View and Layers used for the sensor Figure 8: Isometric View of different sensor layers 127 as RF Interference). One Ground Plane was located on the bottom of the sensor to block any interference coming from the bottom. The top ground plane served as a part of the sensor: as the force applied changes, the top ground plane moves closer to the sensor pad (copper tape) changing the overall capacitance of the sensor.EVA Foam–Dielectric—This was the medium allowing space between the top ground plane and sensor plane to change. This material was chosen because a conventional shoe would have this as its shoe insole providing cushion support on the feet. In a capacitive sensor, the EVA Foam acts as the mechanical element allowing the distance between the two conductive plates to change when force is applied, and rebounds back to its original dimension when the force applied was removed.Copper Tape–Sensor and Routes—This material was chosen as the sensor and the routes due to the ease of cutting in the vinyl cutting machine. There are two layers of copper tape, and both are bonded to a plastic backing sheet. The top layer is for the sensor and the bottom layer is for the routes. This multilayer construction of the sensor is a common practice in flexible PCB Circuit printing to avoid any crosstalk and interference between the routes and the sensor pads. (Jia, 2012). The top sensor layer is contained through a hole so that it could be connected to the routes.Plastic backing–Insulator—This mechanically supported the copper tape sensor and routes during the vinyl cutting process and after the copper has been cut. This plastic backing is chosen due to its flexibility and being tear resistant during the cutting process. This also serves as an insulator and prevents the top sensor pad and bottom route layers come in contact.Polyimide Tape–Insulator—This mechanically supported both the copper taffeta and copper tape by keeping it in place and decreasing the stress on the copper tape’s surface. This was especially useful on the joint when the copper tape was punched through the FCI Clincher, helping to resist the tear on the copper tape. This provided a protective layer from both the copper surfaces preventing them from tarnishing.

### 2.4. Sensor Assembly

The layers introduced above adhered to each other using pressure-sensitive adhesives. Some materials selected already contain a layer of pressure-sensitive adhesive, these being polyimide tape, plastic backing, and copper tape. The bottom copper taffeta is adhered to the plastic backing using polyimide tape. The top copper taffeta was adhered to the cotton cloth (sock liner) by an iron-on fabric glue interfacing. This is a heat-sensitive material that melts and is usually placed between two non-adhesive materials (mainly fabric) to bond them together. The top sensor layer and the bottom route layer were connected using a conductive thread which is secured on its corresponding track route by a piece of polyimide tape (Figure 5). This was then inserted in the through-hole and secured by copper tape on the top sensor layer. The completed sensor assembly was then connected using a 9-way connector to enable attachment to an electronics unit so that it could read the capacitance changes when force was applied. Moreover, this sensor is intended to build inside the shoe. This sensor assembly could adhere to any outsole or midsole of the shoe assembly. 

### 2.5. Sensor Testing and Validation 

#### 2.5.1. VT1: Validation after Sensor Assembly

A bigger sensor was made using the same manufacturing technique as described in the section on sensor assembly for validation of the sensor assembly, To validate the sensor operation, a 50 kg person stood on the sensor and recorded the change in pressure. This was recorded by connecting the sensor to an Arduino with MPR121, this was then connected to visualization software programmed in LabVIEW. These results were compared to a static pressure recorded using an E-med pressure sensing platform. The E-med was considered a gold standard device for measuring plantar pressure. We referred to this validation testing as VT1. 

#### 2.5.2. VT2: Validation for Repeatability and Calibration 

A testing jig was set up with the use of a machined foot phantom so that it can spread pressure evenly throughout the 8 sensors to validate the repeatability of the sensor, Before machining, a deflection analysis was carried out in SolidWorks to see whether three 18 mm panels of MDF would fail under 100 kg of load 100 kg is chosen because a Factor of Safety of 2 is assumed together with a 50 kg body mass. A load simulation was done using SolidWorks Simulation studies, to verify that 54 mm thick of MDF is enough to withstand 100 kg loads. Figure 9 shows that the most displacement was 0.0157 mm. This small deflection ensured uniform load across the sensors during testing.

A compression testing Machine (Hounsfield H10KS) was configured to apply a 100 kg load to apply an accurate load. This machine was programmed and paired to using LabVIEW for Data Acquisition. Figure 10 shows the testing setup for VT2. 

### 2.6. Test Methodology and Test Setup

Two tests were carried out for validating the repeatability of the sensors and for calibration:Loads were repeated 10 times, with the same loading and unloading rate. For this test, the load was repeated 10 times with a single dielectric. The maximum load applied for the compression testing machine was set at 1000 N, as this was an approximate equivalent of a 100 kg mass. The loading and unloading rates were set at 10 mm/min. The sensor was resting on an acrylic sheet to stop it from contacting the metal plate, at the base of the compression testing machine.Loads were repeated 11 times, with different loading and unloading rates. The load was repeated 11 times. This frequency was determined based because it was found on the previous testing, the first test would always be an offset to any remaining tests. The loading and unloading rates varied from 15–50 mm/min with increments of 5 mm/min for each test. A new dielectric was used whenever the loading rate changed and the sensor again rested on an acrylic sheet to stop direct contact with the metal plate base (Figure 11).

## 3. Results

### 3.1. VT1: Validation after Sensor Assembly

A visual comparison was made between the results from the e-med pressure platform Figure 12, which is gold standard equipment for measuring pressure, and LabVIEW results, Figure 13. The E-med result contains more resolution due to it having more pixels and so the foot shape can be made out. The LabVIEW results are depicted as a line graph with 8 lines indicating each sensor pad. The mid-level pressure changes the most as the child grows.

The E-med and LabVIEW results were comparable to each other. The heel areas on both results show similar indications. The heel area on the E-med results shows a large red area, which means that there is a high concentration of force in that area. Similarly, the LabVIEW results show that the heel area has the highest peak, indicating a high relative capacitance reading. This means that the highest concentration of force is also in the heel area.

Although the E-med result shows a large red area on the heel, this does not mean that it is the highest force read. The E-med result also shows a magenta color located on the hallux and the forefoot area. Similarly, LabVIEW results exhibited the second-highest peaks after the heel readings for the plantar pressure sensor. 

The LabVIEW results only show relative capacitance values and not the amount of force for the corresponding capacitance reading. Testing was performed for the second part of validation testing to further investigate the electromechanical behaviors of material combinations. This would then help to characterize the sensors, resulting in an equation to calculate force from capacitance.

### 3.2. VT2: Validation for Repeatability and Calibration 

#### 3.2.1. VT2.1 Results

The graph in Figure 14 shows the time series plots of the change in absolute capacitance during loading and unloading (loading cycle). The peaks indicate one loading cycle of the sensor. The left part of the peak indicated the loading of the sensor, and the right part indicates unloading. In between these peaks, is an almost horizontal line. This indicates the time it took to reset and start the loading again. 

Figure 14 displays 10 peaks indicating 10 loading cycles and 8 sets of plots, representing the 8 areas in the sensor. Each sensor showed different absolute capacitance values, indicated by the plots not overlapping each other. This is due to the sensors having different areas from each other. The highest reading in the graph indicates the heel sensor which had the largest area, whereas the other areas like the midfoot, forefoot, and toes were twice as small as the heel area. Knowing that sensor areas are dissimilar to each other, results in the next set of tests scaled based on the area so that the sensors were characterized easily.

The graph in Figure 15 shows the capacitance and displacement plots of test 1. There are 8 lines representing each sensor. The hallux and the heel plot are above all other readings. The heel readings may be higher than the other sensors because it has the largest area among the other sensors. The hallux performed like the heel sensor, despite it being a smaller area and from the other end of the sensor assembly. This may be because of non-uniformity in the loading system on the foot shape.

When tests 2–10 were performed, the curves now have relative capacitance readings between 0–6 pF (Figure 16). The test 1 result is disregarded because the data trend obtained by the hallux and heel does not follow the results obtained by tests 2–10 as highlighted in Figure 16.

Comparable results were acquired and shown on the Capacitance and Force Plot (Figure 17), where the hallux and heel plot of test 1 is an outlier (Figure 18). 

#### 3.2.2. VT2.2 Results 

For this part of the results, three data sets were analyzed. Each data set was analyzed in three ways—(a) Curves are fitted into each data set (Force and Displacement, Capacitance and Displacement, Capacitance and Force). Each data set has two curves, one for loading and unloading data; (b) coefficients for the equations of the curve fit in each data set are collated, and their variability was observed amongst the same tests and different load/unloading rates. These are represented as box plots; (c) mean values of each data spread in the box plot were acquired and these are statistically analyzed using linear regression techniques. The equation used to acquire Force and Displacement Coefficients is:*F* = *ad^b^*(1)
where *a* and *b* are coefficients, *F* is force, and *d* is displacement. 

The variability of coefficient values per loading rate showed fluctuating ranges as shown in the spread in Figure 19. In the same figure, coefficient ‘a’ showed broad variability at both loading and unloading rates of 30 mm/min, compared to the rest of the loading rates. The ‘b’ coefficients showed a less consistent range throughout the different loading and unloading rates.

The equation used to acquire Capacitance and Displacement Coefficients is: *C* = *ad*^2^ + *bd* + *c*(2)
where *a*, *b* and *c* are coefficients, *C* is capacitance, and *d* is displacement.

### 3.3. Coefficient Variability for Capacitance and Displacement Curve Fit

The variability of all the coefficients at all loading rates in Figure 20 showed an inconsistent spread of coefficient values. However, the coefficient variability during unloading was more consistent. From 15–35 mm/min unloading rate, coefficients showed narrow variability, indicated by a narrow height of the box whereas, on higher unloading rates between 40–50 mm/min, coefficients showed a larger spread of values.

The equation used to acquire Capacitance and Force Coefficients is:
*C* = *aF^b^*(3)
where *a* and *b* are coefficients, *C* is capacitance and *F* is force.

### 3.4. Coefficient Variability for Capacitance and Force Curve Fit

The coefficient variability amongst both loading and unloading rates are inconsistent in Figure 21, this is indicated by the uneven heights of the boxes. 

The results in Figure 22, Figure 23 and Figure 24 show the relevance of the strain rates in the role of curve fits for Equations (1)–(3). Statistical analysis results were derived from MATLAB, and showed no *p* values less than 0.05, indicating that the loading and unloading rates do not affect the curve fits done on Force—Displacement, Capacitance—Displacement, and Capacitance—Force data sets with statistical significance.

Figure 22, Figure 23 and Figure 24 show the plots of mean coefficient values for both loading and unloading data. There is a difference in the values of slopes between loading and unloading. This is indicated by the parameter ‘m’ (slope of the linear regression) shown in Figure 22, Figure 23 and Figure 24. Additionally, there is an offset between the two linear regression lines indicated by different ‘c’ values (linear regression intercept) in Figure 22, Figure 23 and Figure 24.

As the loading and unloading rates do not affect the coefficients derived from the curve fits, mean values of the coefficients shown in box plots of Figure 19, Figure 20 and Figure 21 were acquired to obtain a suitable set of calibration curves. Figure 25 shows the curve fit graphs obtained using the mean coefficient values for each data set.

Referring to Figure 25 there is a significant difference between the coefficients of loading and unloading parameters. It can be observed in the graph there is space between the loading and unloading curves in all the curve fits. This indicated a hysteretic behavior due to the EVA Foam being an elastomeric material. The force and displacement showed little space between the load and unload curves. These curves crossed over each other at the point when displacement is at 4 mm and the force is at 400 N. This is an artifact of the curve fitting of different data sets. The same cross-over feature is starting to show in higher displacement on the capacitance vs. displacement and force graphs.

Although the best fit was chosen for each loading and unloading curve, the mean values of the coefficients do not necessarily represent the best curve fit for the overall data set. A constraint can be given when carrying out the curve fitting so that the loading curve would always be beneath the unloading curve within the operating range.

Both capacitances vs. displacement and capacitance vs. force showed a large space between the loading and unloading curves, even though unloading lines were expected to overlap and be in the same place as the loading lines for Figure 25. This behavior indicated that the readings from the sensor during loading are not the same during the unloading phase. The hysteretic condition is more evident when capacitance is plotted against force and displacement.

The capacitance vs. displacement curve fits took a quadratic form. A characteristic of this curve is that both loading and unloading curves show a dip in the graph between the displacement of 1 and 2 mm. They then show an almost constant slope with a positive linear relationship from the displacement of 2 to 5 mm (Figure 25). This could reflect the behavior of the EVA foam: when the EVA foam is in the early compression stage, the cell walls of the closed cells are bent but not completely collapsed. The uncompressed thickness of the EVA Foam is 3 mm, although this does not necessarily mean that the allowable travel of the head displacement of the load cell is 3 mm. At 3 mm, the cell walls could have collapsed or completely fractured. Beyond 3 mm, cell walls have crushed together, leaving a non-foam EVA material.

The capacitance and force relationship showed a power function. Both loading and unloading curves in this graph have a distinct shape, where the loading curve is less curved compared to the unloading curve. At low and high force readings the curves converge a point. Visually, this graph contained more area between the two curves depicting higher hysteretic behavior compared to force and displacement curves. This could also be due to the elastomeric behavior of the EVA Foam but may include electrical phenomena as well.

The hysteretic phenomenon exhibited in Figure 25 is usually present when an elastomeric foam is under compression loads. This is due to the densification of the materials. Densification is the collapse of cell walls throughout the materials, causing the load to distribute against opposite cell walls (i.e., cell walls are now touching each other bearing the load) (Gibson, 2005). Figure 26 shows a typical stress strain curve of an elastomer, where in the beginning of the curve, a linear stress strain relationship called linear elasticity is depicted. It is determined by bending of the cell walls of the closed cell. The plateau stage corresponds to gradual collapse of the cell wall, through elastic buckling or plastic yielding. The cell wall collapse is dependent on the material. 

For this investigation using EVA closed cell foam, a similar non-linear phenomenon can be seen in Figure 25. The curve fits and graphs in Figure 25 may not be representative of the whole data set but are a representation of behavior during the loading and unloading stages.

## 4. Discussion

The final prototype for built-in-shoe pressure sensing was achieved through iterative design. Suitable manufacturing techniques were selected and the vinyl cutting technique on copper tape and plastic backing was the cheapest and most reliable option available. VT1 mimicked the pressure sensing characteristic of the E-med pressure platform. While VT2 is comprised of two types of testing, one for repeatability and the other with calibration. VT2.1 showed satisfactory results with 10 consistent peaks when the same load was applied 10 times. VT 2.2 determined the impact of different loading rates on calibration curves. These loading rates represented different landing rates for footfalls. The loading rates did not make a difference, with coefficient values of different parameters not showing a consistent range. Moreover, loading rates also were not found to not affect the sensor readings and were not correlated with each other

Creating this new technology may be the first step forward to addressing issues clinicians, researchers, or shoe designers face in accessing shoe pressure data. The next step will be modeling its use in footwear and user testing with children technical results may not solve immediate problems that parents face, but this research increased foot health and footwear literacy.

This research is limited by design. It presents the feasibility of developing low-cost in-shoe sensor technology. This means it does not present clinicians and researchers with an immediately usable tool. In future research, several changes will support future design. In particular, the collapse of cell walls of the EVA could be modeled to better predict and anticipate the capacitance behavior as it is dependent on force and displacement. However, it presents new knowledge regarding new and innovative technologies that are available that can solve data collection issues in the future.

## 5. Conclusions

For future work, more accurate material modeling to better predict material and sensor properties would be helpful. This includes modeling the return of EVA foam during cyclic loading and unloading. This can be combined with the study of electromagnetic fields of capacitive plates sandwiched between soft materials.

Uniform sensor shapes should be implemented, this would make it easier to calibrate the sensors, as they should all experience the same amount of change during the loading cycle.

## Figures and Tables

**Figure 1 sensors-22-08327-f001:**
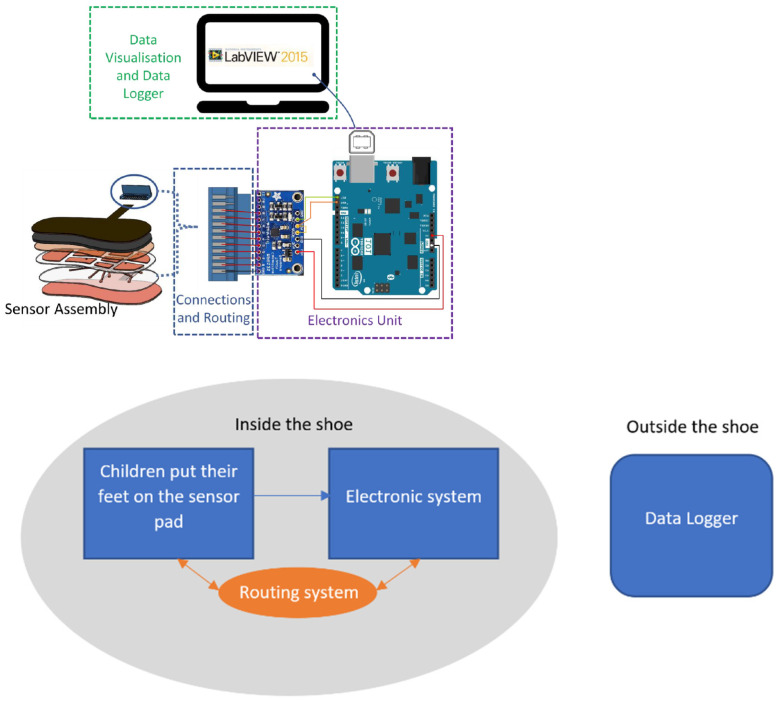
System layout of foot pressure measurement system.

**Figure 2 sensors-22-08327-f002:**
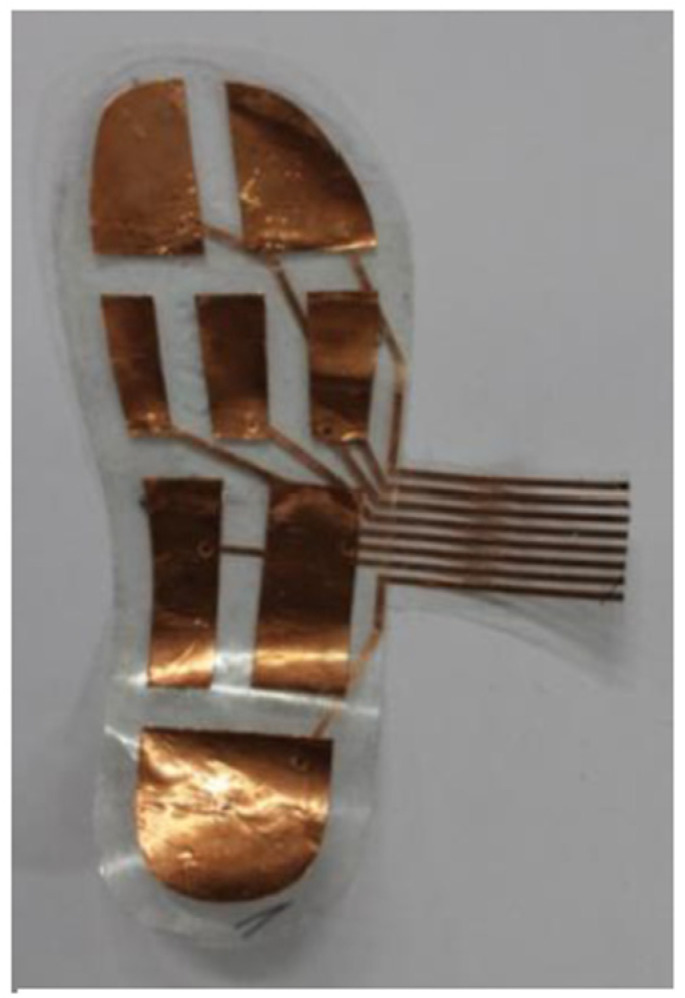
Cut-out of sensor areas using Vinyl Cutter.

**Figure 3 sensors-22-08327-f003:**
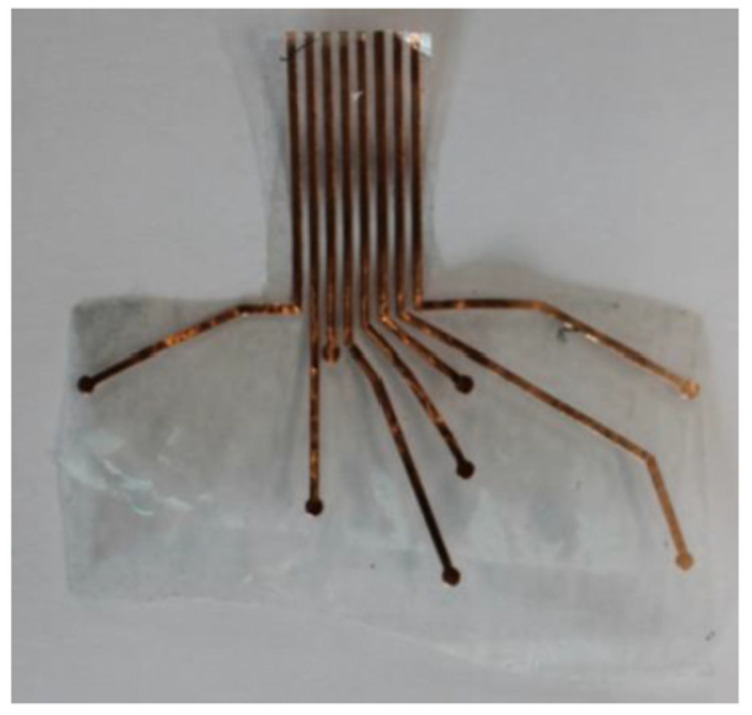
Sensor Route Layer.

**Figure 4 sensors-22-08327-f004:**
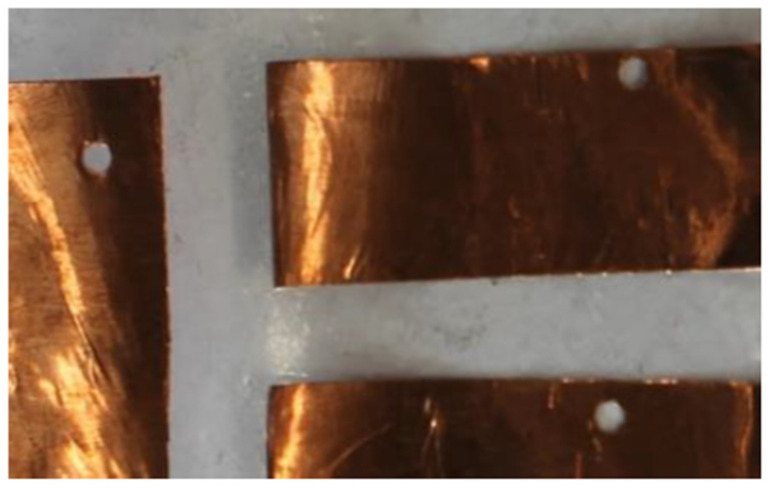
Through holes in the sensor pad for the conductive thread to join with the route layer.

**Figure 5 sensors-22-08327-f005:**
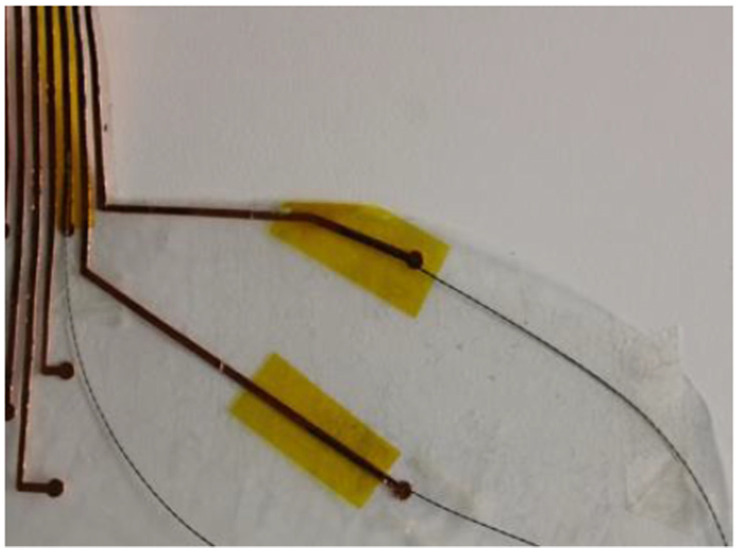
Route layer with conductive tape.

**Figure 6 sensors-22-08327-f006:**
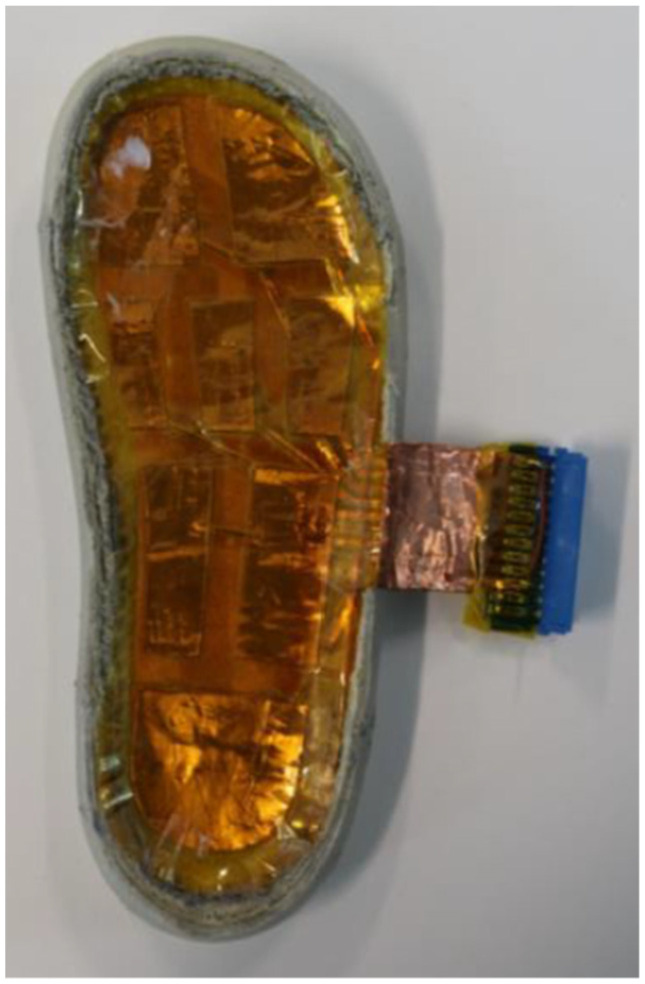
Assembled sensor installed on the outsole.

**Figure 7 sensors-22-08327-f007:**
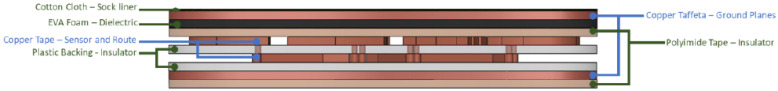
Side View and Layers used for the sensor.

**Figure 8 sensors-22-08327-f008:**
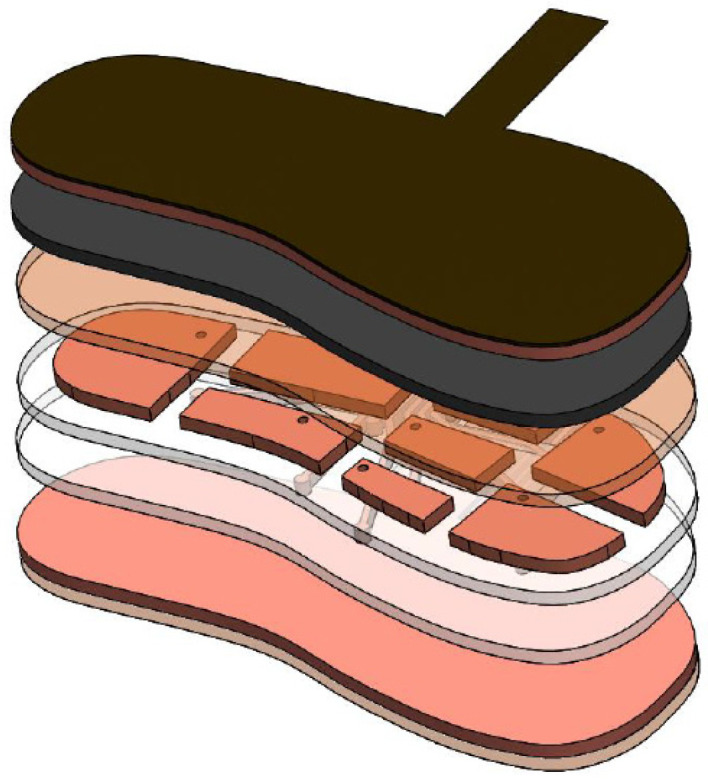
Isometric View of different sensor layers.

**Figure 9 sensors-22-08327-f009:**
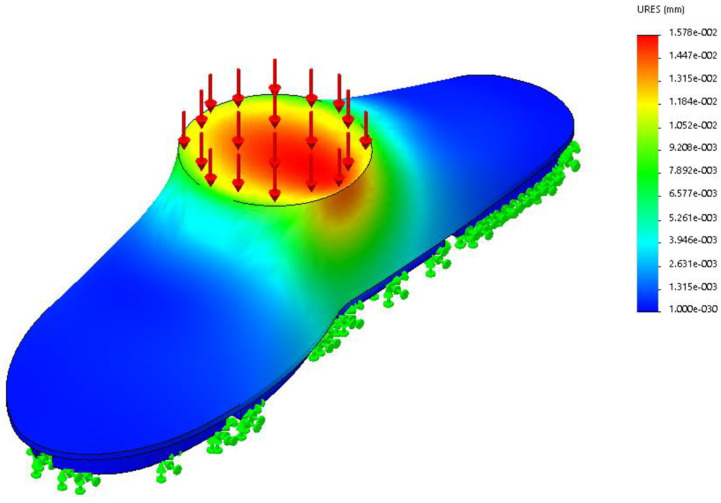
SolidWorks Loading with 100 kg Load.

**Figure 10 sensors-22-08327-f010:**
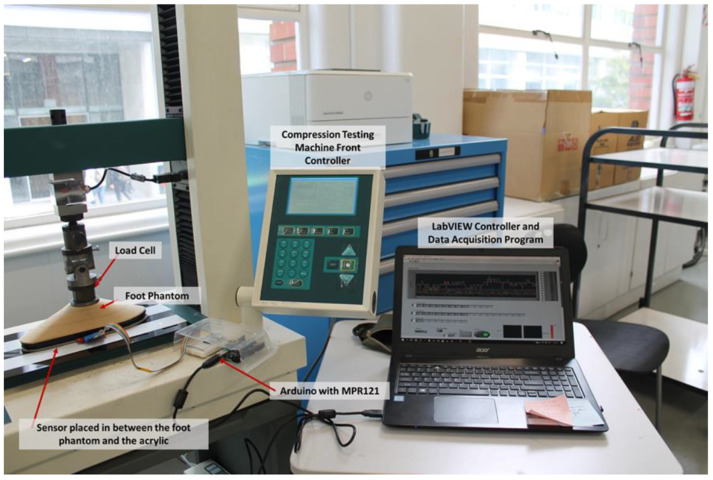
Testing Setup for VT2.

**Figure 11 sensors-22-08327-f011:**
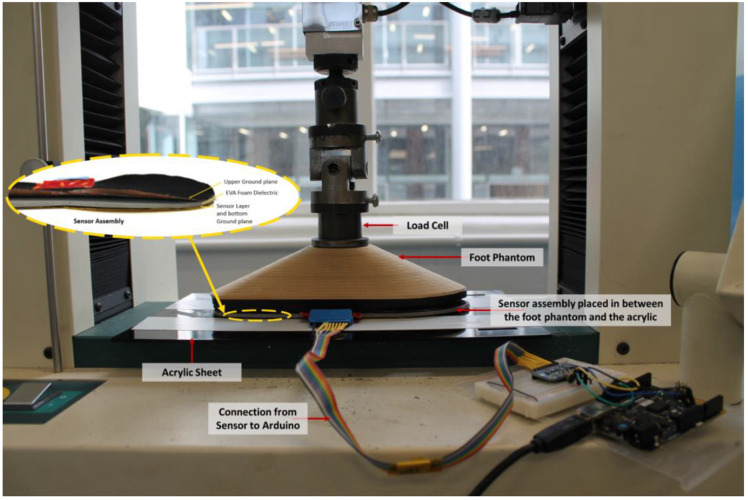
Test Setup for Tests 1 and 2.

**Figure 12 sensors-22-08327-f012:**
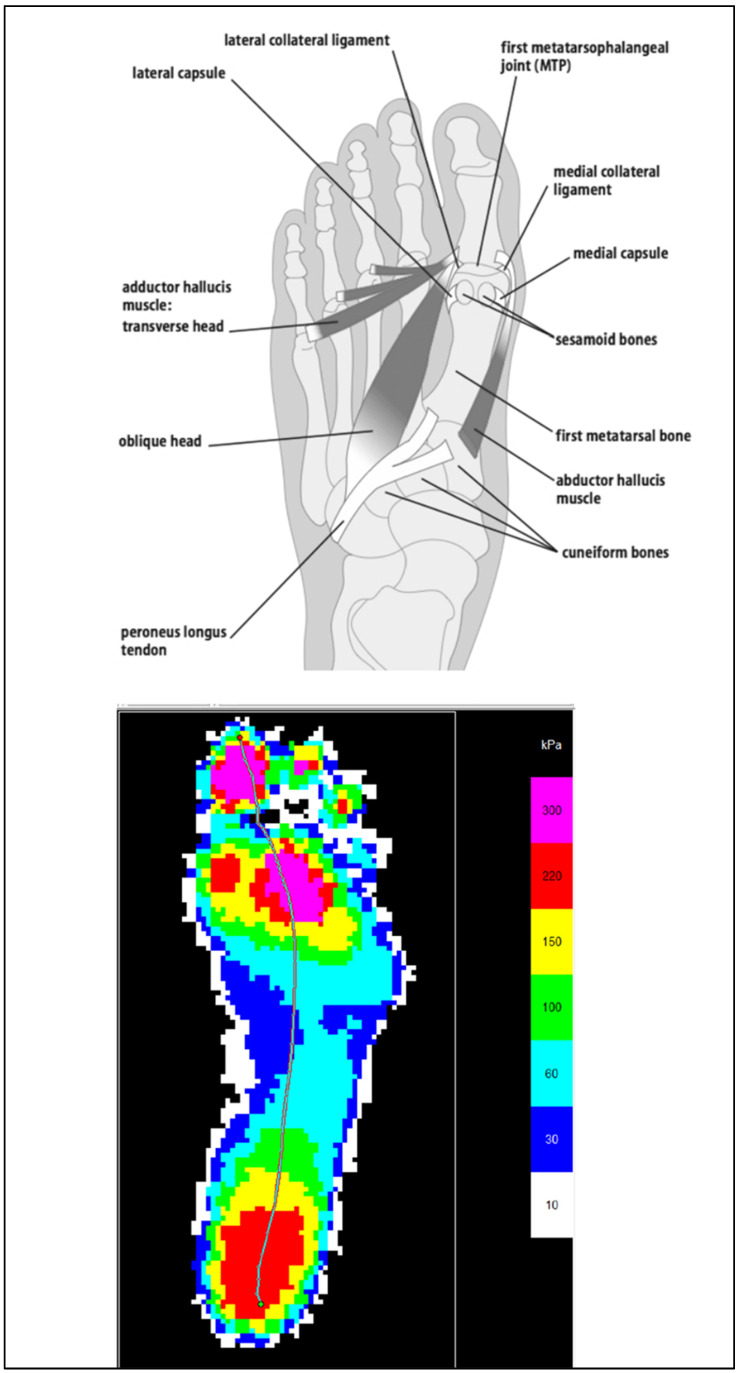
Result from E-med Pressure platform.

**Figure 13 sensors-22-08327-f013:**
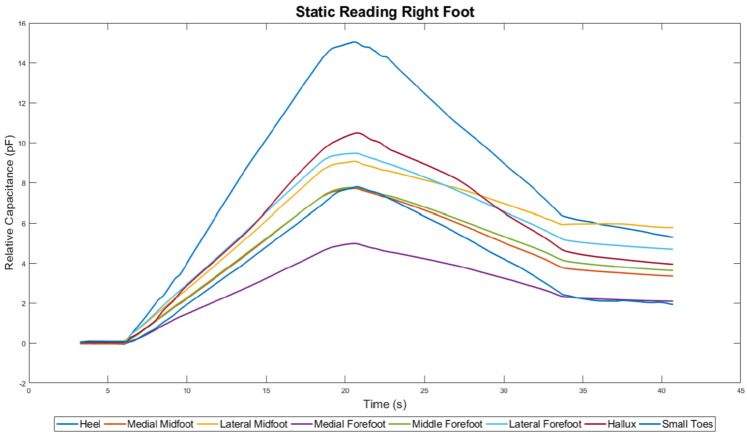
Line Graph of Sensor Prototype Readings from LabVIEW.

**Figure 14 sensors-22-08327-f014:**
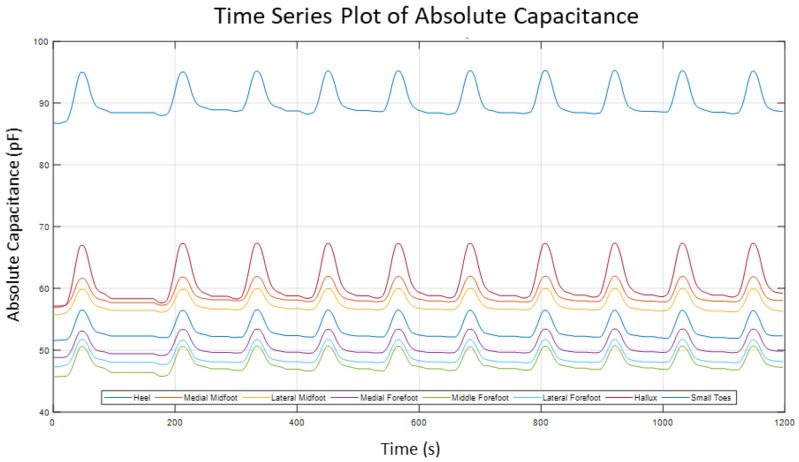
Time series graph of Absolute Capacitance of 8 sensor pads.

**Figure 15 sensors-22-08327-f015:**
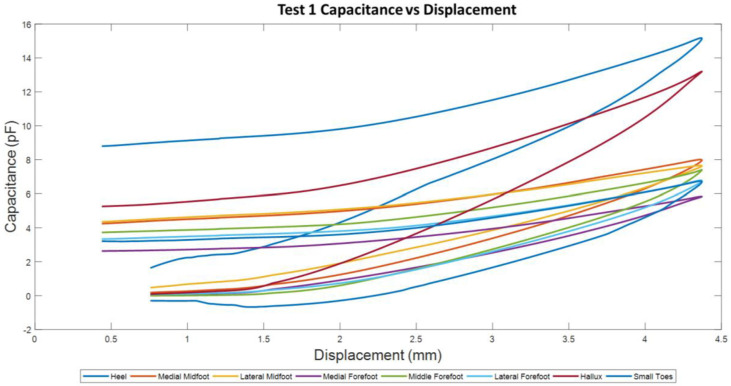
Test 1 Capacitance and Displacement plots for each sensor pad.

**Figure 16 sensors-22-08327-f016:**
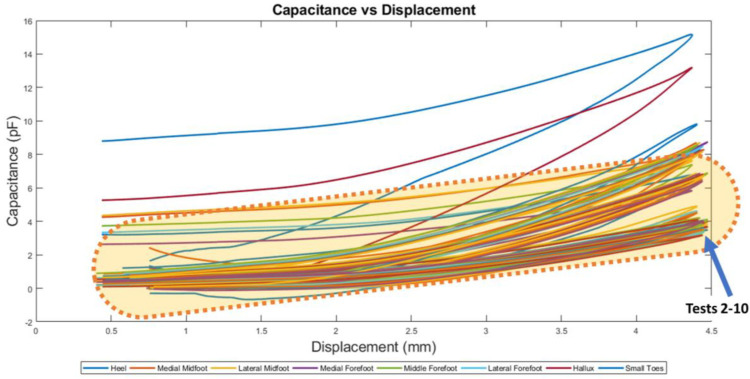
Capacitance and Displacement plots for 10 tests.

**Figure 17 sensors-22-08327-f017:**
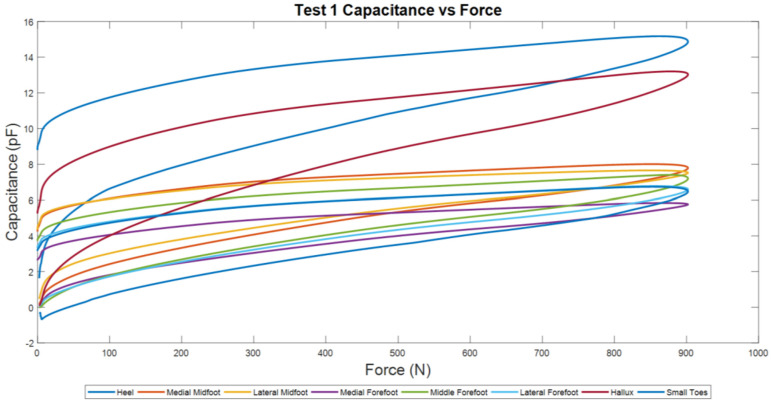
Test 1 Capacitance and Force Plot for each sensor.

**Figure 18 sensors-22-08327-f018:**
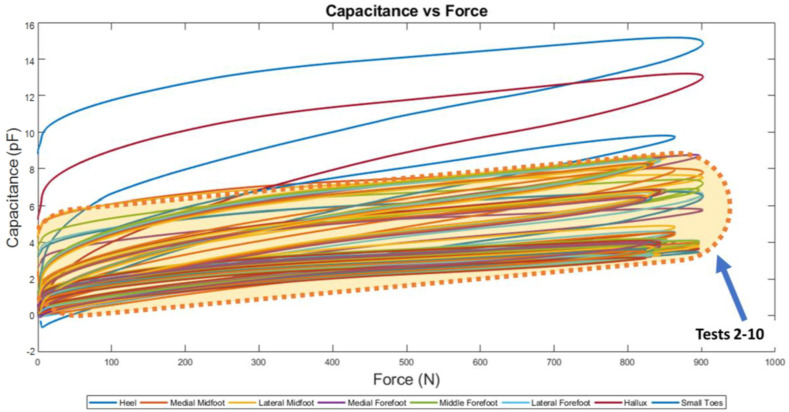
Capacitance and Force Plot for 10 Tests.

**Figure 19 sensors-22-08327-f019:**
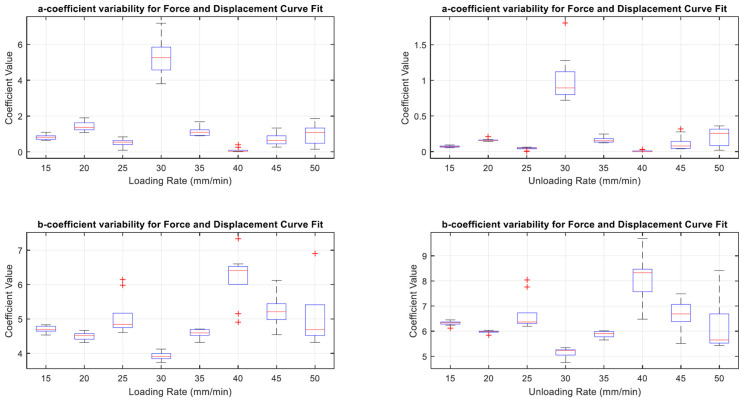
Box Plots showing coefficient variability for Equation (1).

**Figure 20 sensors-22-08327-f020:**
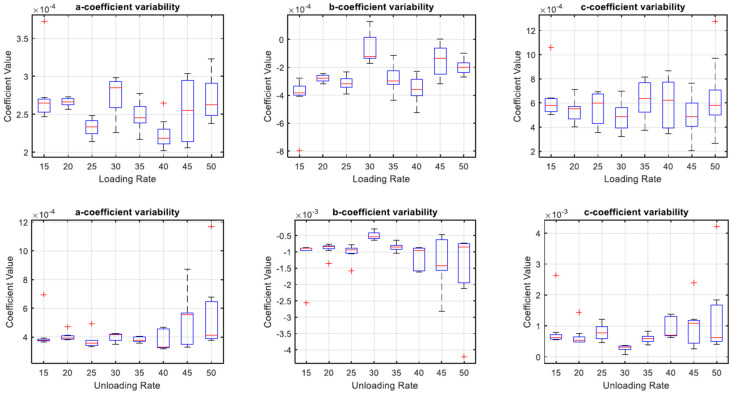
Box Plots of Coefficient Variability for Equation (2).

**Figure 21 sensors-22-08327-f021:**
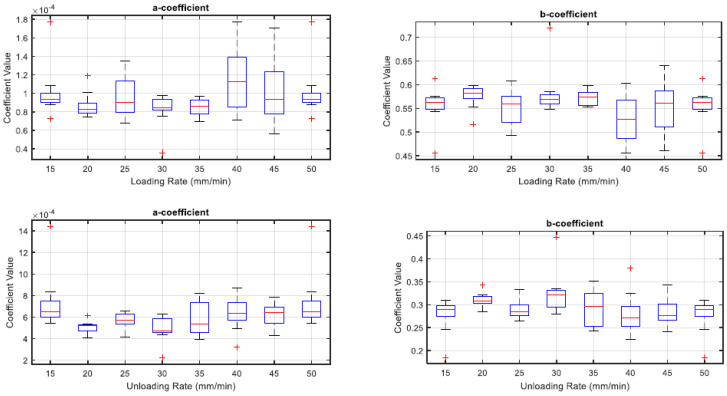
Box Plots for coefficient variability of Equation (3).

**Figure 22 sensors-22-08327-f022:**
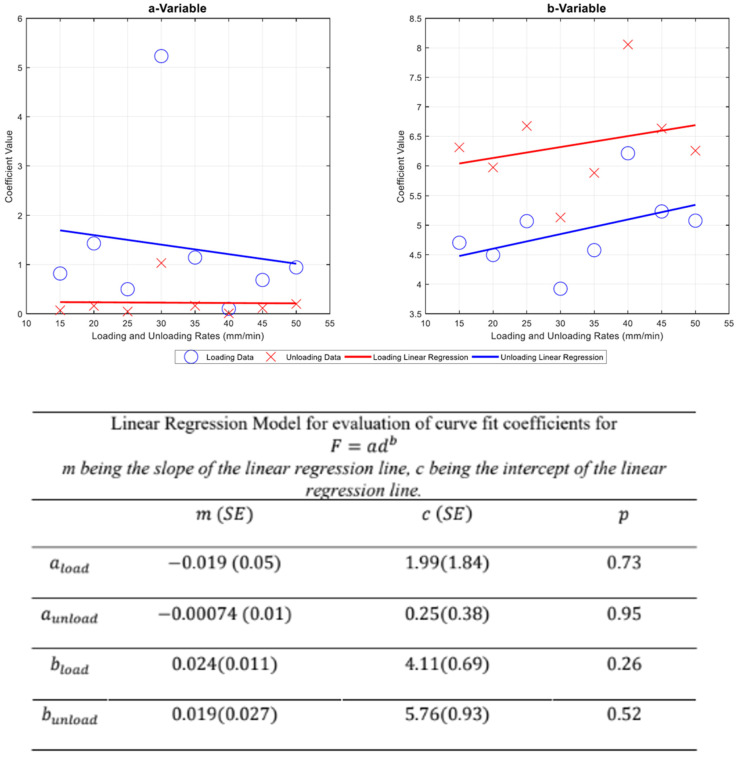
Plot of mean values for Figure 19 with Linear Regression Model Fitted.

**Figure 23 sensors-22-08327-f023:**
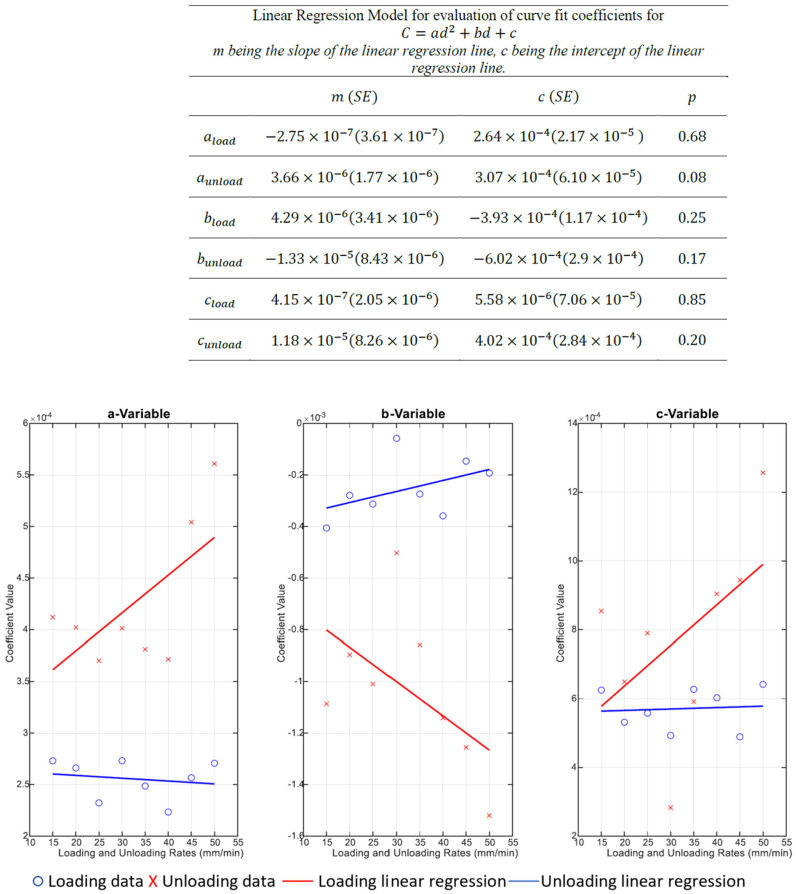
Plot of mean Values from Figure 20 with a fitted linear regression model.

**Figure 24 sensors-22-08327-f024:**
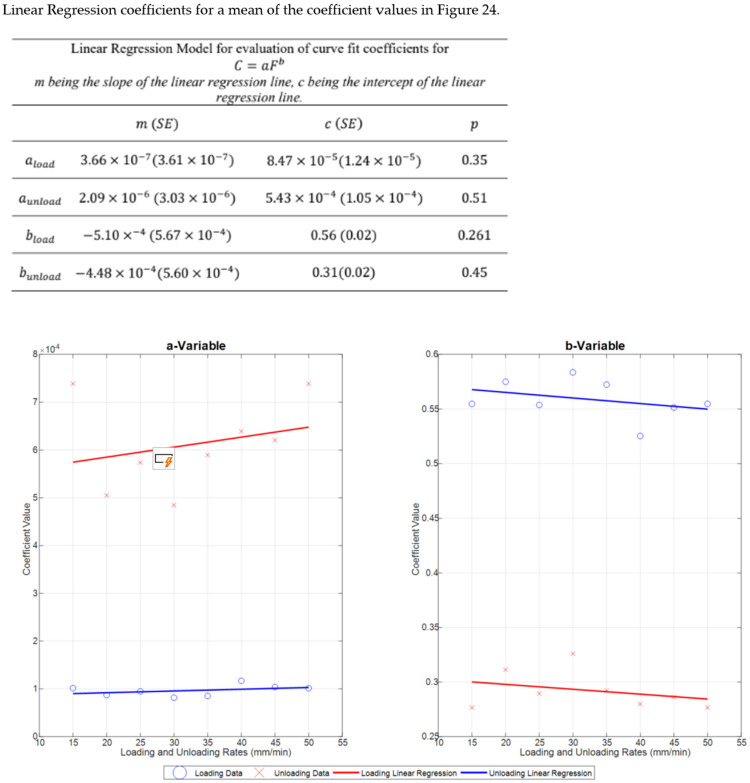
Plot of mean Values from Figure 21 with a fitted linear regression model.

**Figure 25 sensors-22-08327-f025:**
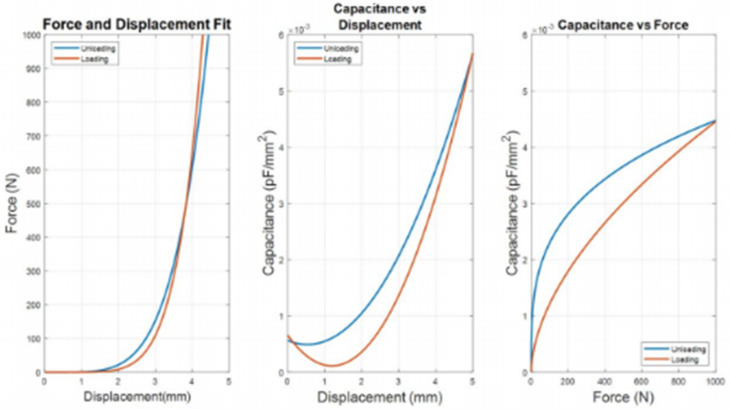
Calibration Curves for different data sets.

**Figure 26 sensors-22-08327-f026:**
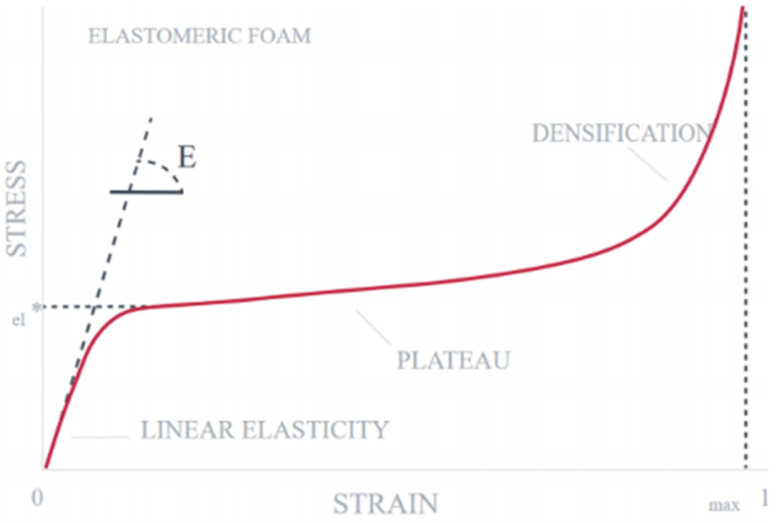
Stress vs. Strain Curve of an elastomeric foam (Carbon (R), 2017).

## Data Availability

Not applicable.

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
