# Peer review of "The Development of a Built-In Shoe Plantar Pressure Measurement System for Children"

_sensors, 2022, doi:10.3390/s22218327_

Round 1

Reviewer 1 Report

The authors developed a low-cost, built-in sole pressure measurement system that detects plantar pressure distribution in young children, and the study aims to improve and provide data on pressure distribution during foot growth. Plantar pressure increases rapidly from infancy to the toddler stage, but little is known about the causes of this change. This emerging technology could be used to help clinicians, researchers, and footwear designers study how pressure distribution varies from infants to young children. However, there are a few factors to modify as a minor issue as for the followings, then it can be accepted to this Journal. The authors should address the following concerns and questions before seeking publication:

1.      On page 3, Figure 1, the corresponding layout can be modified to better reflect the sequence of signal collection and transmission, so that the reader can easily understand the structure and function of the system, the title and keywords of the author's article are related to children, I think it is possible to add the process of children acting on pressure sensing in Figure 1 may better express the meaning of the topic.

2.      On page 4, line 122, “The sensor pads need to be connected to the main electronics unit so that they can be used as pressure sensors. Hence, different manufacturing techniques were explored and qualitatively evaluated. Some techniques are readily available, and these techniques were applied in making the sensors.” It is recommended to cite some relevant papers that use these technologies for sensors, such as:

https://doi.org/10.1002/admt.201800360

https://doi.org/10.1002/aenm.201901124

Sensors 2020, 20, 2925; doi:10.3390/s20102925

https://doi.org/10.1002/admt.202000737

Micromachines 2021, 12, 352. https://doi.org/10.3390/mi12040352

3.      On page 6, line 158,The top ground plane served as a part of the sensor: as the force applied changes, the top ground plane moves closer to the sensor pad (copper tape) changing the overall capacitance of the sensor” I don't quite understand in this sentence why the overall capacitance of the sensor changes when the top ground plane approaches the sensor, I hope the author will explain further.

4.      On page 5, It might be better for the authors to show the variable capacitance in Figure 7 or Figure 8.

5.      There are some typos in the manuscript. Please carefully polish the writing in English.

Author Response

  1. Figure 1 has been modified to provide a better view of the signal collection and transmission process inside and outside the shoe.
  2. Relevant publications have been added to that section of the article now (ref no. 15 and 16).
  3. The capacitance is inversely proportional to the distance between the plates. As the distance between the ground plates and the sensor pad reduces with the application of pressure, the capacitance increases.
  4. Variable capacitance with respect to displacement has been shown in Figure 16.
  5. Grammarly software has been used to fix the typos in the manuscript.

Reviewer 2 Report

What is the hallux region? Perhaps a drawing of the foot showing the different regions would help. Also, where does one expect growth (in the foot) as the child grows older?

How does one expect figure 12 to change as the child grows older? By how much? Is it clear that the sensor described here can detect these changes?

The numbers on the axes of the graphs are too small. Thus, for example, it is not possible to read the times between the several peaks of the graph of figure 14. The same problem holds for other graphs.

It would be good to have data from childrens feet to show how the sensor functions.

Author Response

'Mid-level pressures increase the most as the child grows.' This line has been added on page 11.

Another diagram showing different parts of the foot has been added to figure 12.

E-med platforms are accurate electronic systems for recording and evaluating pressure distribution under the foot in static and dynamic conditions as shown in Figure 12. The results from the sensor are shown on LabView in Figure 13. The heel areas in both Figures 12 and 13 show a similar indication, therefore the sensor is able to detect those changes. The explanation is provided on page 9 - "The E-med and LabVIEW results were comparable to each other. The heel areas on both results show similar indications. The heel area on the E-med results shows a large red area, which means that there is a high concentration of force in that area. Similarly, the LabVIEW results show that the heel area has the highest peak, indicating a high relative capacitance reading. This means that the highest concentration of force is also in the heel area."

Figures 14 to 20 have been made clearer now